# Atomic Force Microscopy of Poliovirus Particles After Inactivation by Chemical Methods and Accelerated Electrons

**DOI:** 10.3390/v17111498

**Published:** 2025-11-12

**Authors:** Sergey V. Kraevsky, Sergey L. Kanashenko, Alena V. Kolesnichenko, Yury Yu. Ivin, Anastasiia N. Piniaeva, Anastasiya A. Kovpak, Aydar A. Ishmukhametov, Sergey V. Budnik, Roman S. Churyukin, Oleg A. Shilov, Dmitry D. Zhdanov

**Affiliations:** 1Institute of Biomedical Chemistry, 10 Pogodinskaya Str., 119121 Moscow, Russiaivin_uu@chumakovs.su (Y.Y.I.); zhdanovdd@gmail.com (D.D.Z.); 2Federal Research and Clinical Center of Physical-Chemical Medicine of Federal Medical Biological Agency, 1a Malaya Pirogovskaya Street, 119435 Moscow, Russia; 3Chumakov Federal Scientific Center for Research and Development of Immune-and-Biological Products of Russian Academy of Sciences (Institute of Poliomyelitis), 8/1 Moskovsky Settlement, Polio Institute Settlement, 108819 Moscow, Russia; 4Teocortex LLC, 34/6 Pervomaysky Settlement, 108808 Moscow, Russiaoshilov@yandex.ru (O.A.S.); 5Department of Biochemistry, Peoples’ Friendship University of Russia Named After Patrice Lumumba (RUDN University), 6 Miklukho-Maklaya St., 117198 Moscow, Russia

**Keywords:** atomic force microscopy, inactivated oral vaccine strains, non-infectious vaccine-like particles

## Abstract

The traditional method used in the production of inactivated vaccines is chemical inactivation using beta-propiolactone or formaldehyde. An alternative method is inactivation by irradiation. Virus inactivation is often accompanied by a change in particle shape, which can negatively affect the preservation of antigens and immunogenicity. Therefore, determining the shape and structure of the viral particle after inactivation is an important step in the development of antiviral vaccines. The poliovirus strain Sabin 2 was inactivated with a dose of 30.5 ± 0.5 kGy. in a pulsed linear electron accelerator with a power of 15 kW and electron energy of 10 MeV. Samples inactivated with beta-propiolactone or formaldehyde were used for comparison. All types of inactivation resulted in D-antigen recovery as determined by enzyme-linked immunosorbent assay. There was no statistical difference between D-antigen recovery in irradiated samples and those inactivated chemically. The shape and structure of the inactivated poliovirus particles were studied using atomic force and electron microscopy. After inactivation with beta-propiolactone or formaldehyde, a change in the native icosahedral shape was observed, with many particles appearing flattened. Specific sorption of antibodies showed that the antigen is mainly preserved in intact capsids for all type of inactivation. However, in the case of inactivation with formaldehyde and accelerated electrons, a significant number of fragments measuring 10–20 nm in height were present. Their proportion was 38 ± 2% and 17 ± 2% for inactivation with accelerated electrons and formaldehyde, respectively. The proportion of bound fragments during inactivation with beta-propiolactone was less than 1%.

## 1. Introduction

Poliomyelitis is an acute infectious disease that occurs only in humans. The causative agents of infection are serologically different polioviruses of types 1, 2, and 3 (PV1, PV2, and PV3) belonging to the Enterovirus coxsackiepol species, Enterovirus genus and Picornaviridae family [1]. Poliovirus is a small non-enveloped virus (approximately 27 nm in diameter) containing an infectious single-stranded RNA of 7441 nucleotides and 60 copies of four capsid proteins: VP1, VP2, VP3 and VP4 [2].

There are two different immunogenic forms of poliovirus, called D-antigen and C-antigen (or H-antigen). The D-antigen form of particles is the native infectious virus and is capable of inducing neutralizing antibodies. C-antigen represents the non-infectious empty or/and immature particles [3]. The conversion of the D-antigen to the C-antigen is induced by numerous environmental stressors, including elevated temperatures (e.g., 60 °C), ultraviolet radiation, desiccation, and exposure to mercury-containing compounds, phenol, or high pH solutions [4,5,6]. Since the production of D-antigen-specific antibodies is an indicator of prolong immune protection against symptomatic poliovirus infection, it was believed that D-antigen is a protective immunogen and, therefore, only the content of D-antigen is crucial for the effectiveness of inactivated polio vaccines [6,7].

Since the establishment of the Global Polio Eradication Initiative (GPEI) in 1988 by the World Health Organization (WHO), the incidence of paralytic polio cases has declined by more than 99%, but the disease is still endemic in a number of countries, such as Afghanistan and Pakistan. Over the past decade, the GPEI has made steady progress towards eradication. Wild poliovirus types 2 and 3 were declared eradicated in 2015 and 2019, respectively; the South-East Asia Region was declared poliovirus-free in 2014; and the African Region was certified as wild poliovirus-free in August 2020 [8].

The GPEI’s immediate goals are to completely interrupt transmission of all wild polioviruses in endemic countries, interrupt transmission of circulating vaccine-derived poliovirus (cVDPV), and prevent outbreaks in non-endemic countries.

According to the GPEI strategy published by WHO in 2021, one of the key areas of activity for research groups around the world is the development of new polio vaccines. In the coming years, in the final fight against the circulation of VDPV and wild strains, the most in-demand vaccine types will be new, genetically more stable oral polio vaccines, the nOPV1, nOPV2, and nOPV3 [9,10], and inactivated vaccines based on Sabin strains [11] as a cost-effective complement to the inactivated vaccines based on wild strains [8].

Over the years, the use of inactivated (IPV) and live attenuated (oral polio vaccine—OPV) vaccines has resulted in a global decline in cases of the disease. Nevertheless, both OPV and IPV have disadvantages that make them suboptimal for use after global eradication. OPV consists of attenuated Sabin strains and stimulates stable immunity. However, through genetic mutations, the vaccine virus can evolve into neurovirulent forms, known as Vaccine-Derived Polioviruses (VDPVs). These VDPVs are capable of circulation in the community and causing outbreaks. The tendency of OPV to genetically revert and give rise to these circulating VDPVs is one of the most significant problems highlighted by the GPEI. The use of nOPVs, genetically more stable variants of the oral vaccines, may lead to a decrease in VDPV circulation due to a reduction in the mutation rate of the new vaccine viruses, achieved through the introduction of genetic modifications into the 3D polymerase sequence and replicative elements, which leads to a decrease in polymerase error and recombination [12]. However, as studies show, the probability of outbreaks of cVDPV (using nOPV2 as an example) is reduced several times, but still not zero [13]. Most IPVs are produced using wild pathogenic strains inactivated with formalin. This production process carries the risk of contamination of personnel, and a violation of the technological process can lead to the release of highly pathogenic strains into the environment. Logically, the ideal vaccine would be an inactivated form of an attenuated strain that can provide protective immunity and simultaneously safely produce the drug [14].

Formaldehyde [15] and beta-propiolactone [16] are widely used chemicals for inactivating viruses in vaccine development. While they effectively reduce virulence and maintain immunogenicity, they are not cost-effective, and are sensitive to contamination during manufacturing. Radiation technologies can mitigate chemical contamination and provide rapid pathogen genome inactivation while preserving immunogenicity [17]. According to radiation theory, the viral genome (RNA in poliovirus) is much more sensitive to radiation damage than the proteins due to their differences in the chemical structure and physical properties (size and density) [18,19]. Accelerated electron inactivation of Sabin poliovirus strains is appealing due to its advantages over other irradiation methods, like gamma or ultraviolet irradiation, without stringent infrastructure requirements [16,17].

Chemical or physical inactivation often results in a significant loss of antigenic activity and immunogenicity due to unpredictable changes in viral morphology or conformational changes in immunogenic epitopes [15,20]. Using formaldehyde for inactivation results in the recovery of 36 to85% D-antigen for all the three types of Sabin strains [21]. Aggregation of viral post-inactivated particles also decreases the amount of active antigen [22]. The study of the morphology of inactivated viral particles can shed light on the nature of the loss of antigen that occurs during inactivation. Before committing to costly immunogenicity studies in animals, candidate inactivated viral particles must be carefully screened. Optimal immune responses are gained through full particle presentation. Therefore, while enzyme-linked immunosorbent assay (ELISA) may identify parts of the capsid that are in the correct conformation, the use of direct imaging techniques may help assess the quality of the antigen being produced rather than just the quantity defined by ELISA-based assays. The combination of atomic force microscopy (AFM) and scanning electron microscopy (SEM) provides insights into the size distribution and morphological changes in vaccine particles [23,24]. Immobilization of polio vaccine particles by monoclonal antibodies to the D-antigen can also indicate the conservation of antigenic sites. In this study, we characterized poliomyelitis virus particles inactivated by formaldehyde, beta-propiolactone, or accelerated electrons using AFM and SEM.

## 2. Materials and Methods

Virus Inactivation and analysis of residual infectivity of virus samples wereperformed similarly to [18,25].

### 2.1. Cells and Viruses

The Vero RCB 10–87 cell line was used as a cellular system for production of the viral material. Samples of the producer culture were prepared from the cell pool of a certified working Vero cell bank (WCBVero, Chumakov Federal Scientific Center for Research and Development of Immune-and-Biological Products of Russian Academy of Sciences, Institute of Poliomyelitis, Moscow, Russia). Attenuated Sabin strains of poliovirus type 1 (strain LSc 2ab, hereinafter referred to as SI), type 2 (strain P712 Ch 2ab, hereinafter referred to as SII), and type 3 (strain Leon 12a1b, hereinafter referred to as SIII) were used to obtain samples and to perform the neutralization reaction. The viral material used in this work was obtained from the original strains by passaging in the Vero cells. The Vero cells for the virus production were cultivated in a bioreactor XDR-50 (Cytiva, Marlborough, MT, USA) using Cytodex 1 microcarriers (Cytiva, Marlborough, MT, USA) in the Eagle MEM medium (Institute of Poliomyelitis) supplemented with 5% fetal bovine serum (LTBiotech, Vilnius, Lithuania). Next, the Eagle MEM medium (Institute of Poliomyelitis) was replaced with medium 199 (Institute of Poliomyelitis) without added serum and the cells were infected with the required strain of the poliovirus. The virus was cultivated at 34 °C until the monolayer of Vero cells on microcarriers was completely degraded, and the viral suspension was collected. The suspension was filtered using filter cascades with a rating (with pore sizes) of 70 μm—0.65/0.45 μm—0.22 μm. The cleared viral suspension was concentrated 200–500-fold using a tangential flow ultrafiltration procedure using a 100 kDa cut-off membrane [26,27].

Hep-2 cells used for virus titer determination were grown in Eagle’s MEM medium (Institute of Poliomyelitis) supplemented with a double set of amino acids and vitamins. The Vero cells used for passage testing for residual virus were grown in DMEM medium (Institute of Poliomyelitis). A total of 5% fetal calf serum (Biolot, Saint-Petersburg, Russia), streptomycin (0.1 mg/mL), and penicillin (100 units/mL) (PanEco, Moscow, Russia) were added to the culture medium.

### 2.2. Virus Inactivation

Polioviral particles subjected to inactivation with formaldehyde, β-propiolactone and various doses of ionizing radiation (accelerated electrons) have been investigated in this work. To inactivate with formaldehyde, poliovirus concentrates were purified using chromatographic procedures (gel filtration and ion exchange chromatography) [28,29] and incubated with 0.025% formaldehyde for 13 days at 37 °C. After inactivation, the samples were subjected to dialysis and transferred to a phosphate-buffered saline solution (PanEco, Moscow, Russia) [26]. To inactivate with β-propiolactone, virus-containing suspension was collected and inactivated with β-propiolactone (1:1000 *v*/*v*) for 48 h at 2–8 °C followed by purification using chromatographic procedures (gel filtration and ion exchange chromatography) [28,29]. To inactivate poliovirus by ionizing radiation, samples with a virus titer of 10^10^ TCID_50_ /mL were used. The samples (3–4 mL) were dispensed into 5 mL cryovials and placed in sealed containers. The samples were irradiated with a dose of 30 kGy at −20 °C on a pulsed linear electron accelerator with a power of 15 kW and electron energy of 10 MeV. Irradiation was performed on one side. The absorbed dose was accumulated in one irradiation cycle (not fractionally). The samples arrived at the irradiation site via a roller conveyor in sealed transport containers; time spent “under the beam” varied from 0.2 s to 2 s. Dosimetric studies were carried out using an absorbed dose detector of an approved type: a standard sample of the absorbed dose of photon and electron radiation (copolymer with a phenazine dye) SO PD (F) R−5/50, passport of a standard sample of an approved type GSO 7865–2000, batch no. 22.57. To determine the absorbed dose, a PE-5400UF spectrophotometer (Ekros, St. Petersburg, Russia) was used. To control the distribution of doses during accelerated electron irradiation, measurements of the absorbed dose were conducted at five points on the sealed containers. The actual measured doses were: 30.1 kGy, 31.1 kGy, 31.0 kGy, 30.1 kGy, and 30 kGy. The average absorbed dose was 30.5 ± 0.5 kGy. The absolute expanded measurement uncertainty, which represents the range around a measurement result where the true value is expected with a 95% probability [30] was determined to be 30.5 ± 3.7 kGy.

The original concentrations for each inactivation process were the same 10^10^ TCID_50_*/*_mL_.

Inactivated by ionizing radiation poliovirus was purified using the same chromatographic procedures. The purity of the purified preparations was evaluated by quantifying residual host cell DNA using real-time PCR and by analyzing protein content using the BCA assay with SDS-PAGE [31]. The residual DNA concentration was below 10 ng/mL in all purified samples, and no contaminating proteins were detected by SDS-PAGE [31].

### 2.3. Analysis of Residual Infectivity of Virus Samples

The effectiveness of viral inactivation was analyzed by titrating the samples on a sensitive Hep-2 cell culture (Cincinnati) obtained from NIBSC (National Institute of Biological Standards and Control, London, UK) according to the 50% Tissue Culture Infectious Dose (TCID_50_) [17], and by conducting a series of passages on a sensitive Vero cell culture and assessing the state of the cell monolayer [21]. To ensure a complete level of inactivation of the viral samples after irradiation with accelerated electrons or chemical inactivation, the 2-step passaging method was used in the Vero cell culture. The sample was diluted 5 times with DMEM to 10 mL and added to a culture flask (25 cm^2^) with Vero cell monolayer and incubated for 5 days at 34 °C. The cell monolayer was inspected using microscope (ICX41, Sunny Instruments, Yuyao, China). Each flask with cell monolayer after incubation was compared with cells not exposed to samples or virus (negative control) and a viral control—cells infected with poliovirus at a concentration of 10 TCID_50_/mL. If there were no signs of degradation, the culture medium from the inspected flask was diluted 5 times in DMEM, added to the 2nd passage cell monolayer of Vero cells, and incubated for 5 days. A sample was considered inactivated if, after a series of two passages in the Vero cell culture, the cell monolayer was found to be intact. A more detailed procedure is described in a previous study [18]. This method is also described in the European Pharmacopoeia and validated. In addition, in the inactivated sample, an infectious titer of the virus should not have been detected by titration in a sensitive culture of Hep-2 cells. These provisions are fully consistent with WHO recommendations for the control of inactivated polio vaccines [32].

### 2.4. Determination of D-Antigen

The D-antigen content of poliovirus samples was quantified using a sandwich ELISA with poliovirus-specific polyclonal rabbit antibodies [33]. Briefly, affinity-purified rabbit IgG, obtained by immunization with purified D-antigen of poliovirus, was used as the capture antibody. The IgG was adsorbed onto a 96-well plate (Corning Costar, Corning, NY, USA) for 20 h at 2–8 °C. Following adsorption, the plate was washed twice with a PBS-T washing solution (0.01 M phosphate-buffered saline, pH 7.4, 0.05% Tween−20) and subsequently blocked with PBS (0.01 M, pH 7.4) containing 1% fetal bovine serum (FBS) for 1 h at 37 °C. The samples, diluted in ELISA buffer (PBS with 0.05% Tween−20 and 1% FBS), were added to the plate and incubated for 2 h at 37 °C. After a second wash step, bound D-antigen was detected using a biotin-conjugated poliovirus-specific rabbit IgG, which was incubated for 1 h at 37 °C. The plate was washed again, and a streptavidin-peroxidase conjugate (Sigma, St. Louis, MO, USA) was added for a 1 h incubation at 37 °C. Finally, the reaction was developed with tetramethylbenzidine (TMB, Sigma, St. Louis, MO, USA) for 15 min, and the absorbance was measured at 450 nm using an iMark microplate spectrophotometer (Bio-Rad, Hercules, CA, USA) [18].

### 2.5. SEM

A drop of 5 µg of the initial solution was applied to a clean hydrophobic surface of a plastic Petri dish. A 3 mm copper grid with a carbon coating (with the carbon film facing the drop) was placed on top of the drop. After holding for 10 min, the grid was washed in distilled water, and the remaining liquid was absorbed with blotting paper. The EM grid was then placed in a vacuum for 2–3 days at a pressure of 5 × 10^−4^ Pa in a Gatan Model 655 setup. Subsequently, the sample was placed in a Hitachi S-5500 scanning electron microscope (Hitachi, Tokyo, Japan) for viewing. Imaging was performed in the transmission mode (BF-STEM).

### 2.6. AFM

AFM measurements were performed using the Dimension Icon microscope (Bruker, Billerica, MT, USA) in the tapping mode in air using the manufacturer’s recommended DAFMCH cantilever holder and RTESPA-150 silicon cantilevers with an elastic constant ranging from ~1.5 to 10 N/m and a resonance frequency of 90 to 210 kHz. Working with the tapping mode in liquid, as the manufacturer’s recommendation, the DTFML cantilever holder and SNL silicon nitride cantilevers with an elastic constant ranging from 0.03 to 0.7 N/m and a resonance frequency of 12 to 80 kHz were employed. To visualize the viral particles using AFM, 5 µL of a suspension of inactivated viral particles was applied on freshly cleaved mica surface. The suspension was allowed to stand for 5 min, and then washed with 1 mL of distilled water and dried with a stream of nitrogen.

Preparation of the specific substrate with monoclonal antibodies was performed with a slightly modified protocol from [34]. Freshly cleaved mica was treated in a glow discharge at a pressure of 2 × 10^−1^ Torr and a current of 100 mA for 2 min. Immediately after activation, 10 µL of protein A diluted to a concentration of 0.5 mg/mL in phosphate-buffered saline (PBSD, Pierce, Waltham, MT, USA), pH 7.4, containing 8 mM NaH_2_PO_4_, 2 mM KH_2_PO_4_, 140 mM NaCl, and 10 mM KCl, was applied on the mica surface for 1 min. The surface was then washed with 100 µL of deionized water. Monoclonal antibodies (IPV ELISA MAbs Types 1, 2, and 3 NIBSC code: 234; 1050; 520) were diluted 1000 times in a similar PBSD buffer and applied to the surface for 10 min. After this, the surface was washed with 100 µL of deionized water and dried with a stream of nitrogen. Inactivated virions were applied as a 5 µL suspension to the surface and kept for 10 min. Non-specifically bound objects were washed for 10 min on a shaker at 300 rpm in the same PBSD buffer. The resulting sample was washed with 1 mL of deionized water to remove buffer salts and dried in a stream of nitrogen.

### 2.7. Statistics

Each experiment was conducted with three independent repetitions. Statistical differences were analyzed by the Mann–Whitney U test using GraphPad Prism 8.0 (GraphPad Software, Inc., New York, NY, USA) software. *p* ≤ 0.05 was considered significant and marked with *.

## 3. Results and Discussion

### 3.1. Virus Inactivation and D-Antigen Recovery

We monitored the residual infectious activity of the treated samples using a two-step passaging method in the Vero cell culture to demonstrate the complete inactivation of the virus. All the treated samples showed no residual activity (Figure 1a,b). The recovery of D-antigen was studied in each inactivated sample to demonstrate its potential immunogenicity. The highest D-antigen recovery rate (76.4–90.0%) was observed for Sabin SI inactivated with formaldehyde or β-propiolactone (Figure 1c). Both SII and SIII demonstrated equal D-antigen recovery within the range of 41.3–45.8% after inactivation with each of these chemicals. The recovery of the D-antigen for SII inactivated with 30 kGy was in the same range (46.0 ± 5.2%) as that for the chemically inactivated samples.

### 3.2. Non-Specific Immobilization

In this work we compared the morphology of inactivated virions of the Sabin polio vaccine strains: SI, SII and SIII inactivated with β-propiolactone and formaldehyde, and the SII strain inactivated with accelerated electrons. Figure 2 shows AFM images of chemically inactivated virions.

According to the published data, the size of virions of different vaccine strains does not differ significantly [35] and is approximately 30 nm. However, in addition to objects of similar size, the images show aggregates and damaged virions, or their fragments, noticeably smaller in size, ~10 nm in height. Below, we will show that such fragments can retain affinity for antibodies.

The formation of aggregates of the formaldehyde-inactivated SIII particles (Figure 2f) can be explained by the fact that the Sabin strain type 3 virion has a different isoelectric point from other types: 6.3 versus 7.2–7.3 for particles of types 1 and 2 [36]. This difference may lead to the observation that particles of different poliovirus types under the same conditions may have varying aggregation abilities due to differences in charges on the surface.

Figure 3 shows the typical electron micrographs of chemically inactivated viral particles, strains Sabin 1, 2, and 3. The EM images confirm the presence of 28 nm virions, their aggregates, and damaged parts of capsids in the samples. Analyzing the gallery of photographs of the viewed samples, it follows that sample preparation, for the purposes of electron microscopy, leads to the formation of a large number of artifacts associated with both the destruction of the original virion during its inactivation, and with the products of the original buffer solution during drying. Therefore, it is important to speak carefully about the differences between the samples based on the EM images.

An AFM image of the Sabin 2 virus particles inactivated by accelerated electrons is shown in Figure 4a. Similarly to the chemically inactivated particles, the height of many particles is less than 28 nm. In addition to spherical particles, triangular particles are also observed. Histograms of the particle height and volume distributions were constructed (Figure 4b,c). The most common particles have a height of 10–15 nm and a volume of about 10,000 nm^3^. This volume corresponds to that of a spherical particle with a diameter of 28 nm. It can be inferred that damaged virus capsid particles may unfold on the mica surface.

### 3.3. Specific Immobilization

Aggregation of viral particles is an important indicator, as it can signal a decrease in the concentration of active antigens [10]. In work [20], an increase in the average size of SARS-CoV-2 particles by one and a half times was observed using the Dynamic light scattering (DLS) method after prolonged exposure to β-propiolactone.

The affinity of the studied objects to monoclonal antibodies is demonstrated in an experiment illustrated in Figure 5. After extensive washing of the sample, only specifically bound capsids and their fragments containing the antigen remain on the surface, while most fragments that have lost antigen are washed off (Figure 5d).

The size distribution of bound particles changed significantly (Figure 6). As expected, intact capsids 28 nm in height were primarily bound to the specific surface. However, in the case of inactivation with formaldehyde and accelerated electrons, a significant number of fragments measuring 10–20 nm in height were present. Their proportion was 38 ± 2% and 17 ± 2% for inactivation with accelerated electrons and formaldehyde, respectively. The proportion of bound fragments during inactivation with β-propiolactone was less than 1%.

The dissociation constant (K_D_) and affinity constant (K_A_) for chemically inactivated Sabin strain type 2 with specific antibodies were measured using the surface plasmon resonance method in [37]. The stronger interaction was observed between SII inactivated using β-propiolactone and MabSII: K_D_ = 5.13 ×10^−11^ M, K_A_ = 1.95 × 10^10^ M^−1^ than between SII inactivated using formaldehyde and the same Mab SII, K_D_ = 5.91 × 10^−10^ M, K_A_ = 1.69 × 10^9^ M^−1^. Apparently, the damaged fragments have a lower affinity to Mab compared the whole capsid, but it is not completely lost.

Unlike formaldehyde [38], β-propiolactone inactivates viruses primarily by altering their nucleic acids [39,40]. This explains why fragments damaged by β-propiolactone inactivation showed virtually no binding to antibodies: β-propiolactone, unlike formaldehyde, does not create random cross-links that could “glue” capsid fragments together, preserving the conformation of epitopes. Most likely, the damage leading to fragmentation is an irreversible breakdown of the capsid, which destroys the conformational epitopes. β-Propiolactone is often considered the preferred agent for inactivation of viral vaccines due to better preservation of antigenicity and more complete inactivation with lower toxicity [20], however, in our study, virus inactivation with β-propiolactone did not show any advantage in preserving D-antigen.

Despite the presence of D-antigen in the viral particles inactivated by accelerated electrons, additional immunogenicity experiments are needed to determine the feasibility of using this inactivation method in vaccine development. Furthermore, it is worth noting that AFM may slightly underestimate the heights of soft biological objects due to the cantilever’s interaction with the viral particle and the effect of dehydration while imaging in air [41].

Note that during long-term storage (more than one month +4 °C), specific agglomerates larger than 1 μm, visible under an optical microscope, are found in the sample inactivated by accelerated electrons (S3).

A morphological study of inactivated vaccine particles of Sabin strains using AFM and EM provides insights into their structural changes during the development and quality control of inactivated vaccines. As demonstrated, inactivation with formaldehyde leads to viral particle aggregation, particularly for the SIII strain. This is consistent with the well-known mechanism of formaldehyde action, which involves cross-linking the amino and imino groups of proteins and nucleic acids [42,43]. Such cross-linking can occur both within a single viral particle, causing its stabilization, and between different particles, leading to the formation of aggregates. Differences in the surface charge of particles at the pH of the buffer used directly influence the electrostatic interactions between them, predisposing some strains to aggregation greater than others [44]. Therefore, the most plausible explanation for particle aggregation is the difference in the isoelectric points of particles from different strains. Aggregation is an undesirable phenomenon, as it can mask antigenic sites and reduce the effective concentration of immunogenic particles. This is indirectly confirmed by the higher K_D_ values for interaction with monoclonal antibodies in viruses inactivated with formaldehyde compared to those inactivated with β-propiolactone [37].

Inactivation of viruses by accelerated electrons is attracting interest as a contactless method that does not require subsequent removal of chemical reagents. During inactivation by accelerated electrons, the viral genome is the primary target for damage leading to loss of infectivity, while the capsid proteins are significantly more stable. This phenomenon is explained by fundamental differences in the structural and physicochemical properties of nucleic acids and proteins. The viral genome is a long, continuous polymer chain. Damage to one strand of the genome at a critical location is sufficient to inactivate the virus. Nucleic acids are more sensitive to the action of free radicals generated during irradiation, primarily hydroxyl radicals. The sugar-phosphate backbone is easily attacked by radicals, leading to chain scission. Irradiation of purine and pyrimidine bases causes their oxidation, hydroxylation, and dimer formation, which blocks transcription of poliovirus RNA. In contrast, the poliovirus capsid protein consists of multiple identical subunits organized into a four-dimensional structure. Damage to the amino acids of one or even several subunits will not necessarily lead to complete destruction of the entire protein shell, and the antigenic function will be preserved [45].

The key observation of our study is the particles with a height of 10–15 nm but a volume equivalent to an entire virus. This convincingly demonstrates that damaged capsids do not fragment, but rather unfold on the mica surface [46]. Ionizing radiation is known to cause breaks in polymer chains (both RNA and capsid proteins), which can lead to loss of structural integrity and flattening of the particle on the mica. The flattering of the particles was observed in our study after samples irradiation. The presence of D-antigen in such particles indicates that key conformational epitopes are preserved even in partially denatured virus particles.

In our study, we have demonstrated that the inactivation of polioviruses Sabin strains by chemicals or accelerated electrons leads to structural rearrangement of viral particles and partial denaturation, while the D-antigen remains preserved.

This work can shed light on the understanding of structure-function relationships for inactivated viral particles and highlights the importance of comprehensive physicochemical and immunochemical analysis in the development and monitoring of antiviral vaccines.

## 4. Conclusions

Inactivated poliovirus particles were characterized using AFM. It has been shown that preparations based on poliovirus inactivation contain not only whole particles, but also their fragments and agglomerates, indicating the potential for optimizing inactivation conditions.

The use of monoclonal antibodies has allowed us to identify a set of viral particles and their fragments that maintain the ability to interact with antibodies generated through various inactivation methods. For the type 2 Sabin strain of poliovirus, only β-propiolactone inactivation resulted in full-size particles interacting with antibodies. However, the formation of smaller fragments with this treatment suggests that these fragments may have lost their ability to bind to antibodies and potentially their immunogenic properties, though further immunogenicity testing is needed for confirmation. Electron irradiation and formaldehyde treatment lead to the accumulation of fragments with antibody affinity, indicating the preservation of specific epitopes despite virion “destruction” which also requires additional testing. It is possible that formaldehyde and accelerated electron treatment may be more suitable for preserving epitopes for binding to virus-neutralizing antibodies, despite potential particle integrity disruption.

## Figures and Tables

**Figure 1 viruses-17-01498-f001:**
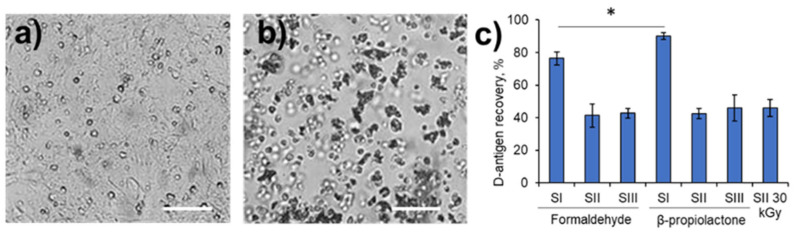
Residual infectivity and D-antigen recovery of the treated samples. Representative micrographs of the Vero cells. The monolayer (**a**) remained intact after incubation with the inactivated poliovirus sample or after incubation with the negative control. The monolayer (**b**) was fully destroyed in the viral control (infection with poliovirus) or after incubation with a non-inactivated poliovirus sample. Scale bars in the images correspond to 250 μm. D-antigen (**c**) recovery in Sabin SI, SII or SIII inactivated chemically with formaldehyde or β-propiolactone, and in SII inactivated with accelerated electrons at a dose of 30 kGy. The results are shown as mean ± standard deviation. *, *p* ≤ 0.05.

**Figure 2 viruses-17-01498-f002:**
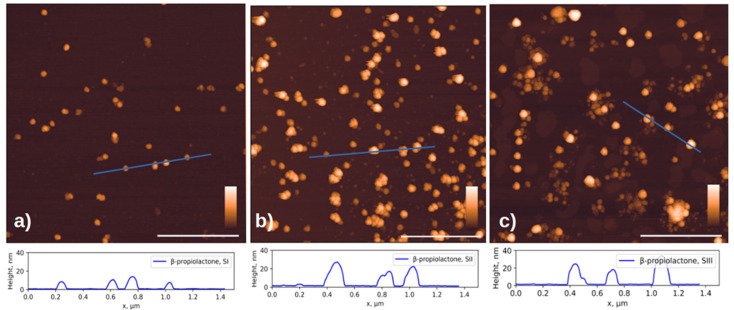
AFM images of three Sabin strains after chemical inactivation. Viral particles inactivated with β-propiolactone (**a**) for SI, (**b**) for SII and (**c**) for SIII; and formaldehyde (**d**) for SI, (**e**) for SII, and (**f**) for SIII. The cross-sectional profiles drawn along the blue lines in the corresponding images above are shown at the bottom of the figure. The white scale bar in the images is 1 μm; the color scale is from 0 to 30 nm for (**a**–**e**) and 0 to 50 nm for (**f**). Additional AFM images available in Appendix A.

**Figure 3 viruses-17-01498-f003:**
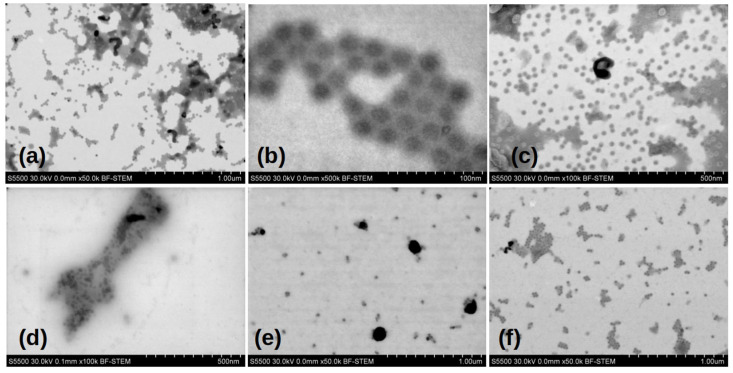
Electron microscopic images of Sabin strains SI, SII, and SIII inactivated chemically. Viral particley inactivated with (**a**–**c**) β-propiolactone; (**d**–**f**) formaldehyde respectively.

**Figure 4 viruses-17-01498-f004:**
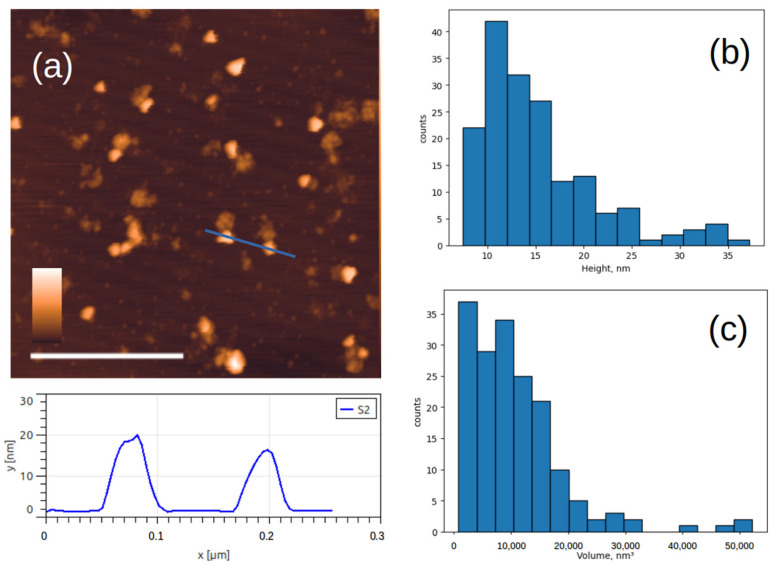
AFM images of viral particles after inactivation with accelerated electrons. (**a**) Typical AFM image of Sabin 2 virus particles and below the cross-section profile drawn along the blue lines; (**b**,**c**) histograms of the height and volume distribution. The white scale bar in the AFM image is 1 μm, and the color scale is from 0 to 30 nm.

**Figure 5 viruses-17-01498-f005:**
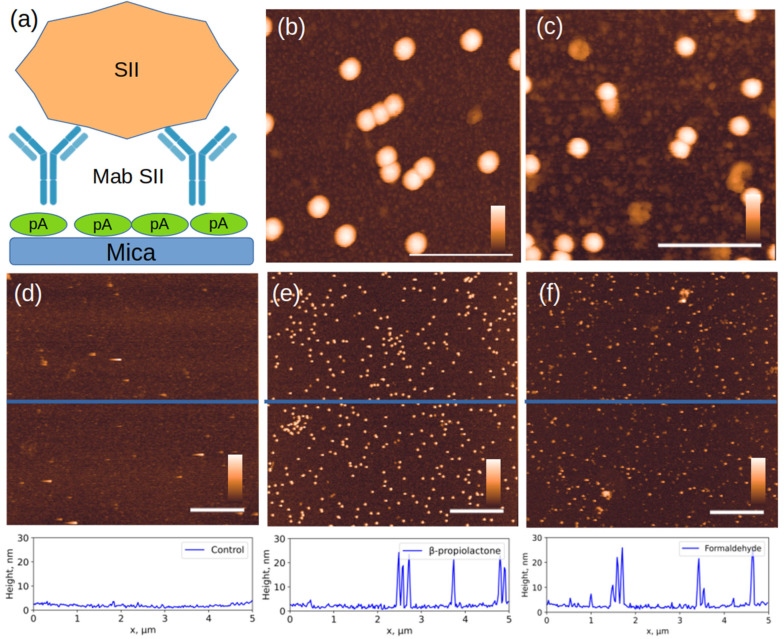
AFM images of objects specifically bound to Mab to the SII strain. (**a**) experimental scheme, (**b**,**e**) SII strain inactivated with β-propiolactone; (**c**) SII strain inactivated with formaldehyde; (**f**) SII strain inactivated with accelerated electrons, (**d**) control sample: inactivated SI on the same Mab SII surface. The section profiles drawn along the blue lines in the corresponding images above are shown at the bottom of the figure. The white scale bar in the images in (**d**–**f**) is 1 µm; in (**b**,**c**), 150 nm. The color scale bar is from 0 to 30 nm. Additional AFM images available in Appendix A.

**Figure 6 viruses-17-01498-f006:**
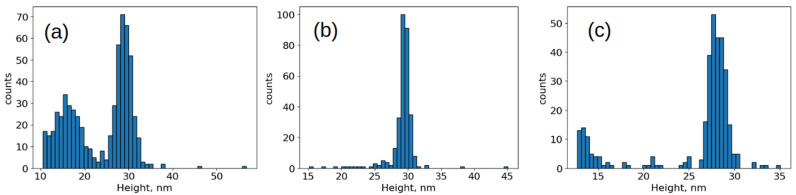
Histograms of the distribution of object heights on a specific substrate. SII strain inactivated by: (**a**) accelerated electrons, (**b**) β-propiolactone, (**c**) formaldehyde.

## Data Availability

The data underlying this research can be obtained from the corresponding author upon reasonable request.

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
