# Peer review of "Atomic Force Microscopy of Poliovirus Particles After Inactivation by Chemical Methods and Accelerated Electrons"

_viruses, 2025, doi:10.3390/v17111498_

Round 1
Reviewer 1 Report (Previous Reviewer 1)
Comments and Suggestions for Authors
Kraevsky et al, set out to assess the potential for accelerated electrons as an inactivation method for poliovirus vaccines in comparison to chemical inactivation in the form of formaldehyde or beta-propiolactone. Whilst the authors have made further progress on addressing the issues highlighted in the previous reviews, there are still a few issues that need to be addressed before this manuscript can be accepted for publication.
Major Points
The abstract has not been updated since the first submission and as the manuscript now contains new data, the abstract no longer accurately reflects the data presented and should be updated.
Line 89-91 – Could the authors please explain their statement, as there is only a single genome packaged per virion, whereas there are 180 copies of capsid protein. Do the authors mean there are a greater number of nucleotides?
Line 103 – The false-positive argument against the use of ELISAs is quite weak. Maybe the authors could rephrase this to highlight that optimal immune responses are gained through full particle presentation, therefore whilst ELISA may recognise parts of the capsid which are in the correct conformation, the use of direct imaging techniques may help address the quality of the antigen being produced rather than just the quantity being defined by ELISA-based assays.
Figure 1c and associated methodology – This figure is an improvement to the manuscript, however the methodology and figure legend do not describe the use of D-antigen specific antibodies in this ELISA, both steps of this ELISA describe a poliovirus-specific rabbit IgG. The capture antibody is described as polyclonal, therefore is likely to bind D and C antigen, and as there is no reference to serotype specific D antigen antibodies for the detection, this assay may just be highlighting how much intact virus particle is remaining following treatment rather than antigen specific.
Discussion – The authors only summarise the results and do not discuss the context of these results within the wider literature nor the comparison between the different inactivation methods described in this manuscript.
Minor Points
Line 47 – The authors refer to the factors that can contribute to C Antigen conversion, with (heat, etc). The term etc needs to be removed and a fuller explanation of what causes the antigenic change and the consequence of this change i.e does C antigen induce antibodies? Are they neutralising/protective? If they ae produced, are they durable in comparison to those induced by D antigen?
Line 48 – The production of D antigen-specific antibodies is an indicator of long lasting immune protection against symptomatic poliovirus infection, rather than indicator of illness recovery.
Line 65 – The authors should remove ‘so-called’ as they are named nOPV-1, 2 & 3. Additionally, the authors should reference the appropriate papers describing these important works in the field.
Line 72-77 – The authors should rephrase to more accurately describe the reversion of OPV to virulence as the term ‘transforms’ isn’t truly reflective of this process. Additionally, the description of the reduction in cVDPV when using nOPV2 is not clear to the reader and should be expanded upon to highlight the improvement of nOPV2 upon OPV2’s reversion rate.
Author Response
Please see the attachment.

Reviewer 2 Report (New Reviewer)
Comments and Suggestions for Authors
To develop antiviral vaccines, this study compared the efficacy of electron beam and chemical methods in inactivating poliovirus. After electron beam inactivation (30 kGy), the virus particles showed higher morphological diversity, but retained their icosahedral shape and D antigen activity, and were able to bind to specific antibodies. In contrast, the inactivation of β - lactone or formaldehyde can cause the particles to flatten, reduce their height, and damage their natural structure. The research results indicate that electron beam inactivation has more advantages in maintaining the structure of viral antigens.
This manuscript has some practical value, but further clarification is needed. My suggestions and comments are as follows:
1, In the ABSTRACT, specific analysis indicators of atomic force microscopy (AFM) and scanning electron microscopy (SEM) in the study (such as standard deviation of particle size distribution, proportion of intact particles, etc.) should be supplemented, and the statistical difference in D-antigen retention rate between 30 kGy accelerated electron inactivation and chemical inactivation should be explained.
2, In the INTRODUCTION section, after elaborating on the goals of the Global Polio Elimination Initiative (GPEI), it is necessary to more clearly connect the "necessity of developing new inactivated vaccines" with the research object of this manuscript, namely the Sabin strain inactivation method, to avoid logical gaps.
3, In the MATERIALS and METHODS, Section 2.1, it needs to specify the model of the bioreactor used for Vero cell culture, the amount of Cytodex 1 microcarrier used, and the method for detecting the purity of the virus after concentration.
4, In the MATERIALS and METHODS, Section 2.2 "Actual absorbed dose is 30.4 ± 3.7 kGy" requires additional results of dose distribution uniformity testing, such as dose differences at different irradiation positions, to ensure consistent inactivation effects.
5, In the RESULTS and DISCUSSION, Section 3.1, it is necessary to supplement the statistical analysis of D-antigen recovery rate (such as analysis of variance, P-value) to clarify whether the differences between different inactivation methods are significant; Figure 1 needs to indicate the calculation method of the error line. Antigen Recovery Rate "76.4-90.0% (Sabin 1 Chemical Inactivation)" requires specific repetition times to be supplemented.
6, In the RESULTS and DISCUSSION, Section 3.2, when analyzing the phenomenon of virus particle aggregation, it is necessary to combine the relationship between the isoelectric point of Sabin 3 virus and the pH value of the buffer solution to explain the specific mechanism of charge difference leading to aggregation, rather than just stating the phenomenon.
7, All quantitative experiments (such as D-antigen recovery rate and particle size measurement) must specify the number of repetitions and indicate whether independent experimental validation has been conducted to ensure reproducibility of the results.
Round 2
Reviewer 1 Report (Previous Reviewer 1)
Comments and Suggestions for Authors
Kraevsky et al, have presented a much improved version of their manuscript and following the minor edits suggested below, the manuscript is now ready for publication.
Minor Points
Line 56-67 – The authors refer to H-antigen impacting immune modulation, however there is no evidence that H-antigen particles have any impact on protective immunity. I suggest the authors remove this statement
Line 102-103 – Whilst I appreciate the changes made, the authors have still not clarified the properties that make the genome more sensitive to radiation damage. Which structural and physical properties are you referring to?
Figure 1c and associated methodology – I appreciate the efforts to present the references in the response to the reviewers, however the authors should reference the studies in which this capture antibody has been characterised as D-Antigen specific, this will be highly important for the readership to assess this work fully.
Author Response
We appreciate the reviewer’s comments. In the following response, we address each calling for changes. For reviewers’ convenience, all corrections are tracked in red color in the main text file. We thank the reviewer for the comments and suggestions.
Comment: Line 56-67 – The authors refer to H-antigen impacting immune modulation, however there is no evidence that H-antigen particles have any impact on protective immunity. I suggest the authors remove this statement
Response: We have deleted this sentence as recommended.
Comment: Line 102-103 – Whilst I appreciate the changes made, the authors have still not clarified the properties that make the genome more sensitive to radiation damage. Which structural and physical properties are you referring to?
Response:
We have discussed this issue in the discussion section. Lines 414 – 427 in the revised manuscript.
Comment: Figure 1c and associated methodology – I appreciate the efforts to present the references in the response to the reviewers, however the authors should reference the studies in which this capture antibody has been characterised as D-Antigen specific, this will be highly important for the readership to assess this work fully.
Response:
The ability of used antibodies to bind D-antigen has been shown in the recent work.
Zyrina, A.; Shishova, A.; Tcelykh, I.; Levin, I.; Shmeleva, O.; Borisenko, N.; Ermakova, M.; Ivanov, S.; Kovpak, A.; Va-silenko, V.; et al. Generation of Polyclonal Antibodies Against Sabin Poliovirus D- and H-Antigens and Their Application in ELISA. Vaccines 2025, 13, doi:10.3390/vaccines13101022.
This research has been cited in the manuscript.
Reviewer 2 Report (New Reviewer)
Comments and Suggestions for Authors
The author responded to the comments one by one, and the quality has been improved. I have no further opinions or suggestions.
Author Response
Thank you for considering our manuscript.
This manuscript is a resubmission of an earlier submission. The following is a list of the peer review reports and author responses from that submission.
Round 1
Reviewer 1 Report
Comments and Suggestions for Authors
Kraevsky et al, set out to assess the potential for accelerated electrons as an inactivation method for poliovirus vaccines in comparison to chemical inactivation in the form of formaldehyde or beta-propiolactone. Whilst this study provides some interesting data in isolation, this manuscript requires significant experimental and editorial changes before being considered for publication. Below are a number of suggestions which, if addressed, would improve the manuscript.
Specific comments
Major Points
Introduction – There is very little introduction to poliovirus itself as a virus and the disease that it causes. Furthermore, there is little information on poliovirus capsid structure, which is integral to this research. The authors need to add significantly more details describing poliovirus and its virions, especially with regards to its antigenic structures, as D antigen particles are only mentioned in Line 61, with no context of explanation as to the importance of this conformation.
Line 39 – The authors state that the poliovirus vaccines are becoming less effective, which is a misleading statement. These vaccines are highly effective and have seen the reduction of paralytic polio cases by over 99.9% globally since the introduction of the GPEI in 1988. I suggest the authors re-write their introduction highlighting the benefits and challenges of poliovirus vaccination more carefully.
Line 48 – Why is this in bold and could the authors please explain their statement, as there is only a single genome packaged per virion, whereas there are 180 copies of capsid protein. Do the authors mean there are a greater number of nucleotides?
Figure 1 – The sizing data produced from the AFM images is biased towards where the authors chose to draw the cross-sectional lines. Different levels of aggregates or disassembled particles are to be expected, for example the ~10nm samples are likely to be capsid pentamers, however the authors do not highlight this nor does the data presented by AFM necessarily reflect the data presented in Figure 2.
Figure 1 and Figure 2 – The authors do not show representative images of virus preparations which have not been inactivated for comparison, these images are necessary for the accurate comparison of the inactivation methods and their impact on particle morphology.
Figure 3 – Why did the authors only assess Sabin type II for electron-based inactivation?
Figure 5 – Can the authors also use these antibodies to assess the overall antigenicity of the samples post-inactivation and compare this to pre-inactivation steps? Whilst showing that post-inactivation that viral particles can still bind to antibody is useful, a quantification of antigenic particles would provide far greater insight into the potential of electron-inactivation of PV particles.
Discussion – The authors only summarise the results and do not discuss the context of these results within the wider literature.
Minor Points
Line 42 – The authors refer to OPV vaccination leading to cVDPV cases, this isn’t well-described for the reader. Could the authors address this sentence to improve clarity and perhaps highlight the recent use of nOPV2 vaccines which have been used to stop transmission chains of VDPV2 with significantly reduced ability to cause cVDPV in its own right.
Line 56 – Could the authors provide examples of known loss of immunogenic epitopes following vaccine inactivation, ideally specifically for poliovirus.
Line 96 – Could the authors clarify if the same concentration and volume of virus particles was used for each inactivation process.
Line 180 – Figure 2A should read 3A
Reviewer 2 Report
Comments and Suggestions for Authors
- Lack of dose-response relationship: Electron beam inactivation was tested at only a single high dose (30 kGy). Morphological or antigenicity data for low/mid doses (e.g., 5–23 kGy) are not provided. This prevents assessment of whether an optimal inactivation dose exists (balancing complete viral inactivation with minimal structural damage).
- Insufficient sample size and reproducibility: AFM/SEM images are limited (especially for the electron-inactivated group, showing only Sabin type 2). Technical/biological replication details are missing (e.g., independent virus preparation batches, repeated inactivation experiments).
- Lack of quantitative analysis in antibody binding assays: Antibody binding experiments (Figs. 4-5) display only representative images, lacking quantitative statistics (e.g., binding efficiency percentage, fluorescence intensity analysis).
- Missing control groups. No AFM/SEM morphological contrl for non-inactivated virus, preventing direct comparison of structural changes pre/post-inactivation; Comparisons between chemical and electron inactivation may be biased due to sample processing differences (e.g., chromatographic purification used only for the formaldehyde group).
- Insufficient AFM image resolution: AFM images (Figs. 1, 3) lack adequate resolution; particle boundaries are blurred, hindering precise size measurement (e.g., subjective assessment of "flattening").
- Unaddressed artifacts in SEM images: Authors acknowledge "extensive drying artifacts" in SEM images (Fig. 2) yet fail to specify how true structural damage was distinguished from artifacts (e.g., was critical point drying applied?).
- Unverified nature of "triangular particles", "Triangular particles" (electron-inactivated group) were not confirmed as intact capsids via 3D reconstruction or antibody labeling; they may merely be fragments.
- Inadequate antigenicity validation (lack of functional data): Antibody binding only demonstrates residual binding capacity. Functional antigen preservation was not confirmed via in vitro immunogenicity testing (e.g., neutralizing antibody titers in animal models).
- Overinterpretation of D-antigen conclusion: The "D-antigen preservation" conclusion relies solely on monoclonal antibody binding. Other epitopes, particularly conformation-sensitive ones potentially vulnerable to inactivation, were not assessed for damage.
- Insufficient mechanistic exploration. The molecular mechanism behind capsid collapse in chemical inactivation (formaldehyde/BPL) is unexplored (e.g., impact of cross-linking on capsid flexibility); The physicochemical basis for antigen preservation by electron inactivation (e.g., selective RNA damage by radicals vs. protein sparing) is only supported by cited literature, lacking experimental data from this study.
- Incomplete analysis of aggregation phenomenon. ">1 μm aggregates" observed after long-term storage of electron-inactivated samples were not analyzed for composition (antigen content?), cause (pH/ionic strength?), or impact on vaccine stability.
- Insufficient validation of inactivation efficacy. Residual infectivity testing is merely referenced as "similar to [14]". Specific data is missing (e.g., log reduction value, achievement of WHO standard ≥6 log reduction).
- Potential artifacts from AFM sample preparation. Nitrogen drying may introduce false-positive aggregation (freeze-drying or liquid-phase AFM is recommended).
- Unverified antibody immobilization. The antibody immobilization protocol did not verify the coating density or activity of Protein A/Monoclonal Antibody (Mab).
- Lack of statistical analysis. Size distribution histograms (Figs. 3b,c, 5) lack error bars and significance testing (e.g., ANOVA for group comparisons), rendering visual interpretations of "higher diversity" unreliable.
Reviewer 3 Report
Comments and Suggestions for Authors
The authors seek to characterize the morphology and immunogenicity of radiation inactivated poliovirus vaccines in comparison with chemical inactivation. Given the limitations of chemical inactivation and the fact that there is currently no radiation-inactivated virus vaccine licensed for human use, this is a subject of potential interest to vaccine developers. Although I consider the overall experimental approach adequate for an initial study, I find the presented results inadequate for publication. For instance, chemical-inactivation results are shown for all 3 poliovirus types, but not for radiation-inactivated ones. The result of one control experiment is shown, but many other relevant control experiments are either not done or not described. An so on. Finally, there is no meaningful discussion of the results or conclusions. I have summarized some specific points below, and would recommend that the authors refocus the experiments and reconsider overall presentation of the results, limitations and conclusions.
***
Lines 37-42: The first paragraph needs to be rewritten to avoid confusion and misinformation. (1) History cannot lead to the development of vaccines, please clarify (2) effectiveness is a vaccine property determined through clinical trials and there is no evidence that effectiveness of any poliovirus vaccine in current use has significantly changed, (3) side effects do not lead to reduced vaccine effectiveness, (4) “rapid evolution” is subjective and should be replaced with “evolution”, (5) evolution and reversion to neurovirulence are not limited to Sabin strain, and other live virus vaccines such as “novel OPV” also evolve and result in VDPVs albeit at a slower rate.
Lines 48-51: No need for bold typeface. In addition, reference #9 is either not published or incompletely cited. As such, I’m not sure if the cited observations about radiosensitivity of polioviruses are just deduced from “radiation theory” or if they are backed up by data.
Line 57: “Therefore, morphological analysis of inactivated virus particles is essential for developing antiviral vaccines” is technically not correct. In fact, inactivated virus vaccines and other vaccines can be developed with no morphological studies. So perhaps a better justification for AFM and SEM studies is needed.
Line 74: Two different strains are referred to as “SI”. In addition, strain Р712 Ch 2ab is a type 2 poliovirus not a type 1 as indicated.
Line 155: Is there a reason SI and SIII were studied by chemical inactivation but NOT by irradiation? Since irradiation is the main subject of these studies, there is no point presenting chemical studies on any strain that does not have a comparable set of irradiation experiments.
Lines 163-164: “According to the published data, the size of virions of different vaccine strains does not differ significantly [1] ...” The sentence is probably correct, but there is no data about the size of virions of different vaccine strains in the cited reference #1.
Line 168: Again, why study chemically inactivated forms but not irradiated ones?
Lines 170-175: Seems like the authors have no confidence in their SEM experimental method. If the results are not contributory because of technical issues, remove from the manuscript. Otherwise, explain the rationale for SEM data and add a discussion of limitations and/or specific technical issues.
Line 174: Is “… drying ect.” a typo representing “… drying, etc.”?
Line 180: Figure 2a should be 3a. Correct?
Lines 181-183: What is the definition of a particle? Is there an assumption that every "lighter than background" fleck in panel 3a is a virus particle or part thereof? If so, there should be a set of control experiments showing that AFM results are uniformly at 0 scale with a virus-negative control sample run in parallel or in an otherwise identical fashion from beginning to the end. (The same issue with definition of a virus particle affects subsequent Figures with AFM data.)
Line 193: What is DLS?
Line 196: Do you mean Figure 4a?
Line 198: What is a “non-specific capsid”? Do you mean there are non-poliovirus capsids in the preps? In addition, Figure caption indicates that panel 4d is a “control” sample? This is not consistent with panel 4d showing specificity of virus binding as suggested in lines 196-198.
Lines 195-198: This paragraph is the only bit of text discussing what is shown in Figure 4, which is a somewhat arbitrary collection of results including two SII results with propiolactone, one SII result with formaldehyde, one SII result with radiation inactivation, and one SIII control. What is the rationale for this mix-and-match? Were other relevant experiments with SI, SII and SIII not done?
Lines 199-201: Figure 5a is unexpected given how clean Figure 4f looks. Similarly, Figure 5b is unexpected given how irregular particles in Figure 4e are. Judging by particle morphology in Figures 4e and 4f I would have expected the exact opposite of what is shown in Figures 5a and 5b. This is worth discussing. In addition, Figure 5 poses another important question. Is irradiation significantly damaging the capsids resulting in such a wide particle size distribution?
Lines 202-204: This is a potentially significant observation regarding the limitation of radiation inactivated vaccines. However, nothing else in the manuscripts discusses optical microscopy, long-term storage issues, etc.
Lines 217-221: This is woefully inadequate in terms of discussion and/or conclusions.
***
Figure 1:
- The six panels a-f should be reproduced at the same magnification. As it stands, it is difficult to compare these panels given different x-axis scales (white bars) and different y-axis baselines (color scale of 0).
- Why are cross sectional profiles shown for one chemical inactivation method (panels a-c) but not the other (panels d-f)?
- Cross sectional profiles (panels g-i; not labeled) should be reproduced at the same y-axis scale to allow direct comparison.
- Panel i (not labeled; the lower right corner) measures particles taller than ~35 microns but the color scale only goes to 30 microns. Does the color scale go from black to white? If not, please clarify.
- Significant clumping of virus particles in panel f requires a discussion or explanation.
Figure 2:
- All panels should be reproduced at the same scale.
Figure 4:
- Please reproduce similar experiments at the same magnification. This includes panels b-f (white scale bars) and 3 unlabeled panels at the bottom (y-axis scales).
- Panel 4d described as “SIII strain, control sample” needs explanation in the methods and/ore here. Specifically, what type of control? No antibody? No virus? Other?
Figure 5:
- Please reproduce at the same x-axis and y-axis scale to allow direct visual comparison.
Please check the manuscript for typographical errors and punctuation marks.
Reviewer 4 Report
Comments and Suggestions for Authors
In the article the Atomic force microscopy of poliovirus particles after inactivation by chemical methods and accelerated electrons, authors presented the results of the studied methods used in the production of inactivated vaccines. A comparisation between the chemicals using beta-propiolactone or formaldehyde and the alternative irradiation method was presented.
There are some uncleared sentences about the poliovirus strains used in the study and the results.
For reader it is necessary to explaine some information about PV a picornavirus with a small non-lipid-containing virion approximately 27 nm in diameter with the icosahedral capsid structure consists of sixty copies of each of the capsid proteins VP1, VP2, VP3 and VP4, encasing a single strand of messenger sense RNA to understand the information from row 50 “ the viral genome (RNA in poliovirus) is more sensitive to radiation damage than the proteins due to their higher molecular weight and the number of copies in a virion”
In the abstract it is not clear that all three PV Sabin strains were tested with chemicals, there were presented as samples in fact the isolates obtaines after cultivation of Sabin strains on cell culture.
In fact Sabin strains (with a mistake of authors on row 74 for type 2 Sabin strain ( P712 Ch 2ab) wich was reffered as type SI ”) were presented in Material and Methods as strains tested.
What is the difference between Sabin 1, Sabin 2, Sabin 3 and Sabin I , Sabin II, Sabin III.
On row 22 it was mentioned that Sabin 2 was inactivated with a pulse linear electron accelerator, must explained why Sabin 1 and 3 was not inactivated by the same method.
In the introduction it was presented on the row 39 that the relative effectiveness of these PV vaccine (IPV and live atteanuated strains) is due to VAPP and VDPV emergency strains which is not true because only for attenuated Sabin strains these effects appeared.
Row 43 – must be more clear if that formaldehyde and beta propiolactone are widely used chemicals for “inactivation” of attenuating viruses in vaccine development. Must be explained why we need to use the attenuation of viruses first and after the inactivation. It is easier to inactivate the wild strains.
In the explanation of the figure the authors used the name Sabin followed by arabic numerals and in the text the roman numerals.
On row 180 – it is a mistake Figure 2a is in fact 3a, there were not discussion about Figure 4 a, b, c in the text.
Comments on the Quality of English Language
The article must be fully revised and carefully checked
Round 2
Reviewer 1 Report
Comments and Suggestions for Authors
Kraevsky et al, set out to assess the potential for accelerated electrons as an inactivation method for poliovirus vaccines in comparison to chemical inactivation in the form of formaldehyde or beta-propiolactone. Whilst the authors have made some progress on addressing the issues highlighted in the previous review, this manuscript does not contain the necessary controls to compare pre- and post-inactivation antigenicity, even using ELISA rather than microscopy. and therefore, this manuscript cannot be accepted.
Reviewer 2 Report
Comments and Suggestions for Authors
non
Reviewer 3 Report
Comments and Suggestions for Authors
I appreciate the authors’ efforts to address my original comments and questions in the revised manuscript and an accompanying cover letter. Although the manuscript has improved, I continue to have concerns about the write up and the methods, including the continued absence of what I would consider to be essential control experiments, as well as a meaning discussion of the findings (and their limitations) in the context of vaccine development or related issues.
Of note, in the accompanying cover letter, the authors have presented an AFM figure of freshly cleaved mica showing a smooth surface suggesting that this is sufficient control for ALL other experiments. I respectfully disagree. Experiments without an antibody require at least one “reagent” control, and experiments with an antibody additionally require at least one experiment with a “non-specific antibody of similar characteristics”. Without such control experiments, the results are just observations that may or may not lead to specific conclusions.
Lines 72-74: The sentence begins to describe paralysis due to cVDPVs but ends up referring to it as VAPP. If the intent is to describe VAPP, please correct the sentence. If the intent is to describe cVDPV associated paralysis, please remove reference to VAPP.
Lines 101-103: Sentence makes no sense.
Lines 103-107: There are many assertions here implicitly suggesting that AFP and/or SEM can replace various immunological studies. These assertions are not substantiated. Morphology MAY be an aid to decision making in future vaccine development, but that is yet to be proven.
Section 2.1 Cells and Viruses: This section needs to include ALL relevant cell culture methods, but additional cell culture details appear Section 2.3, some of which are not consistent with what is described in Section 2.1.
Sections 2.2 and 2.3: Please pay attention to various abbreviations used to refer to tissue culture infectious dose.
Section 2.3: So residual infectivity for all 3 viruses and all 3 inactivation methods was studies in both Vero cells and Hep-2 cells? If so, what were the results? How often did inactivation methods fail? How often did Vero cells and Hep-2 cells agree? And since you tried 5 different radiation doses (mentioned lines 159-160), what were the results? How did the 5 different radiation doses compare in inactivating polioviruses?
Section 2.4: I could be wrong, but my interpretation of SEM data is that only transmission images are shown. If so, what were the findings with reflection mode scanning? If not, please clarify the two different imaging modes vis-à-vis the results shown.
Lines 204-205: Please specify what monoclonal antibodies were used, and show experimental results when a matching non-specific monoclonal antibody is used.
Section 3.1: I still don’t see the point of showing chemical inactivation results for SI and SII if the SEM results (Figure 2) are full of artifacts and AFM results are only for SII. What is the reader to learn from Figures 1c and 1f for example with respect to OVP3?
Lines 235-242 and Figure 2: So 6 images are shown including “a large number of artifacts” and the authors make the conclusion that chemical poliovirus inactivation results in “destruction of virions” as evidenced by the presence of “damaged parts of capsids”. How do I distinguish “damaged capsids” in Figure 2 from other debris? What evidence is there that the artifacts in Figure 2 are not just a technical issue with the specific EM protocol as opposed to the inactivation step? Did you try other EM protocols? Repeated the experiments? Study radiation inactivated viruses by EM?
Lines 281-287: There is no mention of surface plasmon resonance experiments in the Methods section. And again, it is hard to interpret any antibody study without appropriate control data.
Lines 288-290: There is no mention of storage experiments or optical microscopy in the Methods section. What experiments were done and why? How did radiation inactivation compare with other methods? What is the significance of larger >1-micron aggregates? Does this mean radiation vaccine preparations are not suitable for cold-chain storage and field administration?
Section 5 (or 4?): I appreciate the expanded discussion. It would be nice to include a specific section on limitations of these studies, and put the new findings in the context of existing literature.
Reviewer 4 Report
Comments and Suggestions for Authors
There were made the modification in according with the request.